# The Effects of Different Antigen–Antibody Pairs on the Results of 20 Min ELISA and 8 Min Chromatographic Paper Test for Quantitative Detection of Acetamiprid in Vegetables

**DOI:** 10.3390/bios12090730

**Published:** 2022-09-05

**Authors:** Yuxiang Wu, Yemin Guo, Qingqing Yang, Falan Li, Xia Sun

**Affiliations:** 1School of Agricultural Engineering and Food Science, Shandong University of Technology, Zibo 255049, China; 2Shandong Provincial Engineering Research Center of Vegetable Safety and Quality Traceability, Zibo 255049, China

**Keywords:** ACP, vegetables, antigen–antibody pairs, quantum dots, 8 min test

## Abstract

To establish rapid, high-sensitive, quantitative detection of ACP residues in vegetables. A 1G2 cell clone was selected as the most sensitive for anti-ACP antibody production following secondary immunization, cell fusion, and screening. The affinity of the 1G2 antibody to each of the four coating agents (imidacloprid–bovine serum albumin [BSA], thiacloprid–BSA, imidaclothiz–BSA, and ACP-BSA) was determined using a 20 min enzyme-linked immunosorbent assay (ELISA). The half maximal inhibitory concentration (IC_50_) was 0.51–0.62 ng/mL, showing no significant difference in affinity to different antigens. However, we obtained IC_50_ values of 0.58 and 1.40 ng/mL on the linear regression lines for 1G2 anti-ACP antibody/imidacloprid–BSA and 1G2 anti-ACP antibody/thiacloprid–BSA, respectively, via quantum dot (QD)-based immunochromatography. That is, the 1G2 antibody/imidacloprid–BSA pair (the best combination) was about three times more sensitive than the 1G2 antibody/thiacloprid–BSA pair in immunochromatographic detection. The best combination was used for the development of an 8 min chromatographic paper test. With simple and convenient sample pretreatment, we achieved an average recovery of 75–117%. The coefficient of variation (CoV) was <25% for all concentrations tested, the false–positive rate was <5%, the false–negative rate was 0%, and the linear range of the method was 50–1800 μg/kg. These performance metrics met the ACP detection standards in China, the European Union (EU), and the United States (US). In summary, in this study, we established an 8 min QD-based immunochromatographic stripe for the rapid and accurate quantitative determination of ACP residues in vegetables.

## 1. Introduction

ACP is a highly efficient new generation systemic pesticide that first appeared on the market in the 1990s. This chloronicotinyl compound kills pests on contact or after ingestion. Equipped with a novel insecticidal mechanism, ACP effectively prevents and controls the breeding of pests such as those in the orders Hemiptera, Lepidoptera, and Coleoptera. It is a highly active chemical with quick and long-lasting effects; only a small amount is needed to kill insects. ACP is particularly effective on pests that have developed high resistance to organophosphorus, carbamate, and pyrethroid insecticides. It is soluble in acetone, methanol, ethanol, dichloromethane, chloroform, acetonitrile, and tetrahydrofuran.

As stipulated in the *National Food Safety Standard of China*—*Maximum Residue Limits for Pesticides in Foods* (No. GB2763-2019), the maximum residue limit (MRL) for ACP in vegetables such as tomatoes, napa cabbage, and eggplants is 1 mg/kg, and for vegetables such as radishes and round cabbage, it is 0.5 mg/kg. In Chapter 12 of EU Regulation 396/2005, the MRL for ACP in plant products has been updated to 0.1–0.9 mg/kg, while according to the United States Environmental Protection Agency (US EPA), it is 0.3–2 mg/kg. Chromatography, including gas chromatography (GC), high-performance liquid chromatography (HPLC), gas chromatography–mass spectrometry (GC-MS), and liquid chromatography–mass spectrometry (LC-MS), is the most widely used method at present for ACP quantification. However, it requires expensive instruments and professional handling. Samples must undergo complex preprocessing before measurement, which is time consuming. Chromatography is therefore not suitable for rapid high-throughput detection of samples. References [1,2,3,4,5,6] of this study discuss fluorescence detection methods based on aptamers and aptasensors. The limit of detection (LoD) in spiked serum and water is <200 pm, a level of sensitivity that is indeed better than that of a monoclonal antibody (mAb). However, the detection methods mentioned in the references take a long time to perform, requiring 2–4 rounds of incubation with >2 h total incubation time and involving many experimental steps. Therefore, they are not suitable for onsite rapid quantitative detection.

The 8 min chromatographic paper test for quantitative detection of ACP in vegetables in this paper is based on the combination of highly sensitive antigen–antibody pairs, which is a reflection of the innovation of immunological layer analysis note design. Through screening and pairing, we discovered highly sensitive pairs of complete antigens and mAbs. The four haptens—imidacloprid, thiacloprid, ACP, and thiamethoxam—are highly similar in structure. We coupled them with carrier protein to form complete antigens and tested them against ACP mAb, and we were ultimately able to establish an 8 min rapid quantification test for ACP in vegetables and other types of samples. This study lays a foundation for future efforts towards the fast, efficient, and accurate determination of ACP content in vegetables.

## 2. Materials 

### 2.1. Main Reagents

We used the following reagents in this study: imidacloprid, thiacloprid, ACP, and thiamethoxam (all from TanMo Quality Testing Technology, Beijing, China); tetramethylbenzidine (TMB; Shanghai Ruiyong Biotechnology, Shanghai, China); horseradish peroxide (HRP)–linked goat anti-mouse immunoglobulin G (IgG) and Type 501 water-soluble adjuvant (both from Shandong Lvdu Biotechnology, Binzhou, China); and bovine serum albumin (BSA) and keyhole limpet hemocyanin (KLH; both from Solarbio Life Sciences, Beijing, China). Carboxyl functional ZnCdSe/ZnS quantum dots (QDs; maximum emission, 625 nm) were obtained from Shandong Landu Biotechnology Co., Ltd., Binzhou, China).

### 2.2. Main Instruments

The main instruments used in the study were: a microplate reader (MK3; Thermo Fisher Scientific, Waltham, MA, USA); a constant temperature incubator (DRP-9272; Senxin, Shanghai, China); a high-speed centrifuge (Hettich ROTOFIX 46; Andreas Hettich GmbH & Co. KG, Tuttlingen, Germany); a microscope (DX-2; Shanghai Teelan Optical Instruments, Shanghai, China); a CO_2_ incubator (MCO-18AIC; PHCbi, Sanyo, Japan); a constant temperature shaker (ZWYR-2102C; Shanghai Zhicheng Analytical Instruments, Shanghai, China); a fume hood (SW-TFG-12; Zhejiang Sujing Purification Equipment, Shangyu, China); a nitrogen evaporator (MD200-1; Shanghai Huxi Instruments, Shanghai, China); a low-speed centrifuge (TDL-40B; Anting Scientific Instruments, Shanghai, China); a Sujing clean bench (SW-CJ-2FD; Suzhou Purification Equipment, Suzhou, China); a constant temperature dehumidifier (DP-20S; Dorosin Air, Guangzhou, China); a QD fluorescence detector (LD-01) and an incubator (both from Shandong Lvdu Biotechnology, Binzhou, China).

## 3. Methods

### 3.1. Preparation of Immunogen and Coating Antigen for ACP and Three Other Coating Agents 

We suspended ACP (100 mg) and 50 mg of 3-mercaptopropionic acid in 5 mL tetrahydrofuran, and then we added 10 mL of sodium ethoxide to facilitate the reaction. The reaction mixture was heated under nitrogen protection at 80 °C for 6–8 h and then distilled under reduced pressure to obtain carboxylated ACP hapten B-ACP as the solid crude product [7,8,9]. We prepared the complete antigens ACP-KLH (immunogen) and ACP-BSA (coating antigen) using the conventional 1-ethyl-3-(3-dimethylaminopropyl) carbodiimide (EDC) reaction. The coupling route is shown in Figure 1. We used conventional ultraviolet (UV) absorption spectroscopy for detection as described in the literature [10,11,12,13,14,15]. We repeated this synthesis method three times, and the conjugating rate reached over 80%. The UV had the absorption peak of protein and ACP, which showed successful conjugation. We synthesized three other coating agents (imidacloprid, thiacloprid, and thiamethoxam) following the same preparation procedure, and the coupling route is shown in Figure 1.

### 3.2. Preparation and Affinity of Anti-ACP mAb

We mixed the ACP-KLH immunogen and the Type 501 water-soluble adjuvant at a ratio of 10:1 (v:v) and injected the mixture into the leg muscles of BALB/c mice aged 8–10 weeks (according to the conventional dosages, 50 μg/mouse, 100 μL/mouse). Cell fusion was performed on day 3 after the second immunization. We prepared the identified antibody using the conventional method as described previously by Woychik et al. [16] with minor modifications. Wells showing strong positive results were selected for affinity measurement [17,18,19,20,21,22,23] via indirect competitive enzyme-linked immunosorbent assay (ELISA) using the cell supernatant. For the ELISA, we added 50 μL of 10-ng/mL ACP standard to each well, followed by 50 μL/well of goat anti-mouse HRP-linked antibody. Next, we added 50 μL tenfold-diluted cell supernatant and mixed the solution using gentle shaking. The mixture was allowed to incubate in a dark environment at 37 °C for 15 min. We washed the wells thoroughly 3 times with 250 μL/well wash solution at 10 s intervals and patted them dry with absorbent paper. Next, 100 μL/well of TMB substrate solution was added, with gentle shaking to mix. We covered the plate with cover film and placed it in the dark at 37 °C for 5 min to react. Wells that showed high affinity were selected and subcloned by limiting dilution. We drew the cell supernatant on day 7 and tested it. The ELISA plate was coated with the ACP coating agent ACP-BSA at an optimal concentration of 0.01 μg/mL. ACP standard (0, 0.1, 0.3, 0.9, 2.7, 8.1 ng/mL) was used for indirect ELISA screening. We calculated the binding rate (%) of ACP for the concentrations used:binding rate (%) = B/B0 × 100%(1)
where B0 was OD_450_ without ACP, and B was OD_450_ with ACP. An S-shaped standard curve was plotted with B/B0 as the *y*-axis and log^10^ of the standard solution concentration as the *x*-axis. We plotted logit/log to find the linear regression equation, with logit as the *y*-axis and the log of ACP standard concentration as the *x*-axis, as per the following equation:logit = ln(p/q), p = B/B0, q = 1 − p(2)

### 3.3. Specificity of Anti-ACP mAb

We determined the cross-reaction rate of the antibody with eight nicotinyl substances (ACP, imidacloprid, thiacloprid, imidaclothiz, clothianidin, thiamethoxam, dinotefuran, and dinitridin) by testing antibody specificity via 20 min ELISA, similar to the antibody affinity test described in Section 3.2. The results are shown in Table 1. 

### 3.4. Affinity of Anti-ACP mAb and Four Kinds of Coating Agents in ELISA 

The affinity of 1G2 anti-ACP mAb with three other coating agents (imidacloprid–BSA, thiacloprid–BSA, and thiamethoxam–BSA) was measured using 20 min ELISA as described in Section 3.2. The results are shown in Table 2. 

### 3.5. Application and Performance of Four Coating Agent/Anti-ACP Antibody Pairs in QD-Based Immunochromatography

#### 3.5.1. Performance of the Four Coating Agent/Anti-ACP Antibody Pairs

In accordance with references [24,25,26,27,28,29,30], we covalently bonded the carboxyl functional group on carboxyl-functionalized ZnCdSe/ZnS QDs to the amino group of the antibody–protein conjugate. The carboxyl group was activated by EDC/N-hydroxysuccinimide (NHS) to form an intermediate and the fluorescent antibody. We striped test lines T (the four coating agents at 1 mg/mL: imidacloprid–BSA, thiacloprid–BSA, imidaclothiz–BSA, and ACP–BSA) and a control line C (goat anti-mouse IgG at 1 mg/mL) onto a nitrocellulose membrane using a dispensing instrument at 0.5 μL/cm. The sample pad, fluorescent antibody pad, and absorbent paper were successively assembled onto the PVC card, cut into strips, and inserted into the cartridge to form the detection kit. We added 0, 0.1, 0.3, 0.9, 2.7, and 8.1 ng/mL ACP standards to the sample diluent (0.01-M phosphate-buffered saline (PBS)) to perform quantitative determination following the steps outlined in Figure 2 for the preparatory performance of the four coating agent/anti-ACP antibody pairs. Then, the best combination of coating agent/anti-ACP antibody pairs was used to establish the standard curves. Quantification results are shown in Figure 3.

#### 3.5.2. Sample Pretreatment and Accuracy Analysis

Three sample pretreatment methods were evaluated, as described below.

Method 1: To 10 g of the sample vegetable (napa cabbage in this case; other samples are referred to this method), we added 10 mL of 0.01-M PBS. Vegetable juice was extracted by homogenization and centrifuged at 8000 rpm for 5 min. Then, we drew 10 μL supernatant and added it to 1mL 0.01-M PBS. Finally, we removed 100 μL of the solution and tested it according to the steps in Figure 2.

Method 2: First, the cabbage leaves were cut into small pieces (0.2cm × 0.2cm) with scissors, then, 5g were placed in a beaker, and we added 100 mL of 0.01-M PBS to soak the pieces for 30 min. Finally, we removed 100 μL of the solution and tested it according to the steps in Figure 2.

Method 3: To 1g of the sample vegetable, we added 10 mL ethyl acetate, mixed well by vortexing, and centrifuged the mixture at 8000 rpm for 5 min. Then, 100μL supernatant was transferred to another centrifuge tube and blown dry with nitrogen at 50–60 °C. Next, we added 1mL of 0.01-M PBS to re-dissolve the solid. Finally, 100 μL of the solution was drawn and tested according to the steps in Figure 2.

Then, 100 μL test solution was removed to the sample well of the detection card, and the detection card was placed in the incubator at 37 °C for 8 min. Finally, the card was taken out and put into the QD reader to read the T and C absorbance values and the T/C ratio (excitation wavelength/emission wavelength: 325 nm/625 nm). The QD reader was stored with a linear standard curve of T/C ratio and the ACP standards so the detection content could be presented on the reader display screen, and the paper detection report could be printed.

## 4. Results

### 4.1. Preparation and Specificity of Anti-ACP mAb 

Finally, we screened hybrid tumor cell 1G2 as the best choice because its affinity measured via ELISA was the best (IC_50_ = 0.61 ± 0.07). The 1G2 anti-ACP mAb showed a >50% cross-reaction rate with imidacloprid and thiacloprid, a 20% cross-reaction rate with imidaclothiz, and a ≤5% cross-reaction rate with all other pesticides (Table 1).

The affinities of all four coating agents (all showing >82% cross-reaction rates with the 1G2 anti-ACP mAb) as determined by ELISA are shown in Table 2. We observed no significant difference in half maximal inhibitory concentration (IC_50_) values of the anti-ACP mAb to the four coating agents or ACP.

### 4.2. Application and Performance of Four Coating Agent/Anti-ACP Antibody Pairs in QD-Based Immunochromatography

Figure 3 A–D shows pairing results for the 1G2 anti-ACP antibody with ACP-BSA, imidacloprid–BSA, imidaclothiz–BSA, and thiacloprid–BSA. The added 0, 0.1, 0.3, 0.9, 2.7, and 8.1 ng/mL ACP standard to the six test strips are shown in A, B, C, and D. No obvious color gradient was observed on the chromatogram for the 1G2 anti-ACP antibody/ACP-BSA. Affinity of the 1G2 antibody to imidaclothiz–BSA was too weak to produce a sufficiently dark color for chromatographic detection at the same concentration of 1 mg/mL for T line. Both 1G2 anti-ACP antibody/imidacloprid–BSA and anti-ACP antibody/thiacloprid–BSA showed good color gradients for different concentrations of ACP standard. We also observed a good linear relationship at 0.2–4.9 ng/mL from Figure 3 E. The IC_50_ values obtained from the linear regression lines of 1G2 anti-ACP antibody/imidacloprid–BSA and anti-ACP antibody/thiacloprid–BSA were 1.40 and 0.58 ng/mL, respectively. This corresponded with threefold higher sensitivity of the former *versus* the latter in chromatography. So, the best combination for an 8 min chromatographic paper test for quantitative detection of acetamiprid in vegetables is the 1G2 anti-ACP antibody /imidacloprid–BSA pair.

### 4.3. Sample Pretreatment and Accuracy Analysis

The best combination of 1G2 anti-ACP antibody /imidacloprid–BSA pair was applied to establish the 8 min chromatographic paper test for quantitative detection of ACP in vegetables. As shown in Table 3, the average recoveries for sample pretreatment methods 1, 2, and 3 were all 75–117%, the coefficient of variation (CoV) was <25% for the concentrations used, the false–positive rate was <5%, and the false–negative rate was 0%. We achieved a linear range of detection from 50 to 1800 μg/kg for average samples according to sample pretreatment methods 1, 2, and 3, which well satisfied the maximum limit (1000- μg/kg) stipulated in the GB2763-2019 standard of China, the 100–900-μg/kg limit required by the EU, and the 300–2000-μg/kg limit required by the US. As method 3 requires nitrogen blowing, which takes time to finish, and method 1 needs to be centrifuged, method 2 is the more practical choice because of its convenience.

### 4.4. Comparison with LC-MS Methods 

To further verify the accuracy and reliability of the 8 min chromatographic paper test for quantitative detection of ACP in vegetables, 50 cabbage samples were detected by the 8 min chromatographic paper test and HPLC-MS methods with two repetitions. The result shows that ACP was not detected (below the LOD of 5 μg/kg) in all samples except for sample NO. 6 (65.54 μg/kg) and NO. 37 (105.66 μg/kg) by the LC-MS method. For the 8 min chromatographic paper test, ACP was also not detected (below the LOD of 50 μg/kg in all samples except for sample NO. 6 (80.7 μg/kg) and NO. 37 (118.2 μg/kg). The results of the two methods are essentially in good agreement, indicating that the developed 8 min chromatographic paper test method can be used to detect ACP in vegetables. 

## 5. Conclusions and Discussion

As Type 501 water-soluble adjuvant is directly released into the muscle tissues of immunized mice, it is readily absorbed and stimulates splenocytes to produce antibodies, in contrast with the sustained release of immunogen. Two immunizations were therefore sufficient before cell fusion was performed to produce the high-affinity mAb 1G2.

In this study, we found that in combination with four coating agents (imidacloprid–BSA, thiacloprid–BSA, imidaclothiz–BSA, and ACP-BSA) in ELISA, the antibody produced by the 1G2 cell clone (obtained by ACP-KLH immunogen screening) showed no significant difference in IC_50_ to ACP (all IC_50_ values fell within the range of 0.51–0.62 ng/mL); however, we observed great differences in the chromatographic test strip results. We therefore chose the 1G2 anti-ACP antibody/imidacloprid–BSA pair as the best antibody–antigen combination for the chromatographic kit development. The stark differences between ELISA and chromatographic results might be explained by the following factors. The ELISA reaction takes time to complete, and the antibody was heavily diluted. The antibody–antigen reaction was performed in the aqueous phase in the microwells, allowing adequate reaction time for both the antigen and the antibody. We observed no significant difference in ELISA results among the antibody–antigen pairs. In chromatography, however, the reaction time was short. Both the antigen on line T and the antigen standard were fiercely competing with the antibody in a nearly dry state on the chromatographic strip. As the antibody is produced by immunizing mice with ACP-KLH, it is more likely to bind with the ACP antigen standard, to which it has a higher affinity than with the other three antigens when they are present on line T. This inhibits the binding of the antibody to the antigen on line T and yields a larger color variation.

In addition, this study compared three simple sample pretreatment methods. Ethyl acetate requires nitrogen blowing to remove, which is time consuming. From the perspective of operational convenience, method 2 is more convenient than the others and is the optimal extraction method.

## Figures and Tables

**Figure 1 biosensors-12-00730-f001:**
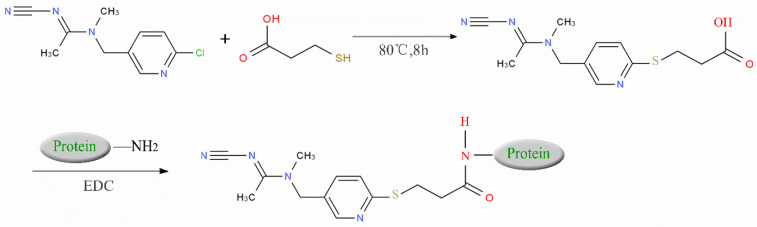
ACP coupling to form complete antigen.

**Figure 2 biosensors-12-00730-f002:**
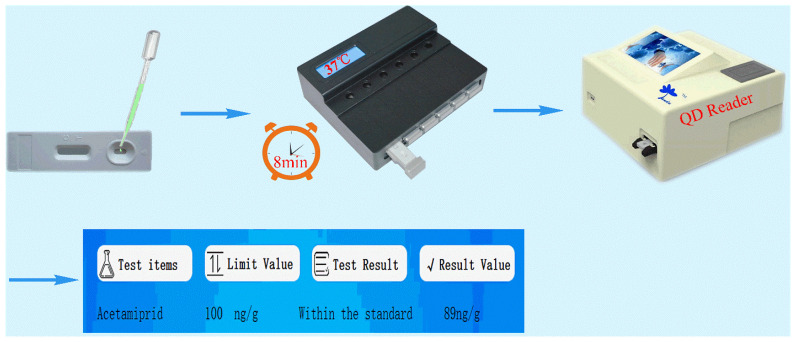
Eight-minute immunochromatographic strip quantification of ACP.

**Figure 3 biosensors-12-00730-f003:**
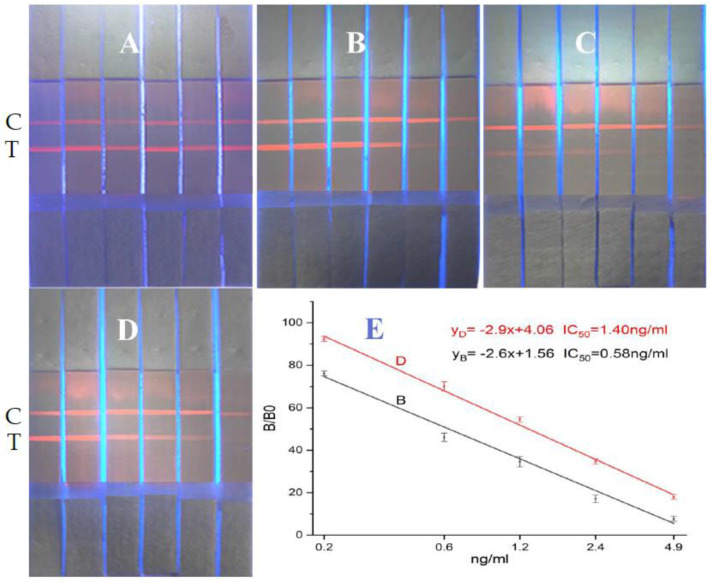
QD-based immunochromatographic results of the four coating agent/anti-ACP antibody pairs. (C represents control line, T represents the detection line. Four kinds of conjugated antigens ((**A**): ACP-BSA; (**B**): Imidacloprid–BSA; (**C**): Imidaclothiz–BSA; (**D**): Thiacloprid–BSA) were coated at the concentration of 1 mg/mL; 0, 0.1, 0.3, 0.9, 2.7, and 8.1 ng/mL of ACP standards from left to right. (**E**) D represents the standard curve for 1G2 anti-ACP antibody/imidacloprid–BSA pair; (**E**) B represents the standard curve for anti-ACP antibody/thiacloprid–BSA pair).

**Table 1 biosensors-12-00730-t001:** Specificity of clone 1G2 anti-ACP mAb.

Chemical	Structural Formula	IC50 (ng/mL)	Cross-Reaction Rate (%)
ACP	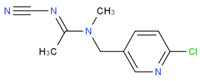	0.61 ± 0.07	100%
Imidacloprid	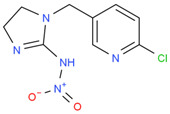	0.72 ± 0.06	83%
Thiacloprid	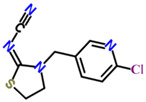	1.07 ± 0.09	56%
Imidaclothiz	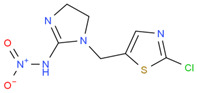	3.21 ± 0.26	20%
Clothianidin	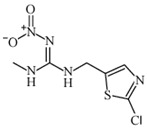	12.1 ± 1.32	5%
Thiamethoxam	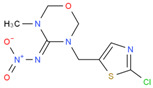	30.2 ± 3.36	2%
Dinotefuran	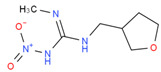	60.1 ± 6.29	1%
Dinitridin	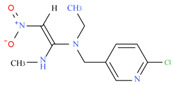	100 ± 11.09	0.6%

**Table 2 biosensors-12-00730-t002:** ELISA results for specificity of anti-ACP mAb with each of the four coating agents.

Coating Agent	Coating Agent Concentration	IC_50_ (ACP)
ACP-BSA	0.01 μg/mL	0.62 ng/mL ± 0.06
Imidacloprid–BSA	0.03 μg/mL	0.51 ng/mL ± 0.05
Thiacloprid–BSA	0.07 μg/mL	0.55 ng/mL ± 0.05
Imidaclothiz–BSA	0.3 μg/mL	0.58 ng/mL ± 0.05

**Table 3 biosensors-12-00730-t003:** Comparison of sample pretreatment methods and determination of accuracy.

Sample Pretreatment Method	Concentration Used (μg/kg)	Concentration Measured (μg/kg)			Average Recovery (%)	Coefficient of Variation (%)
		1	2	3		
Method 1	10	10.6	9.5	9.1	97.3	5–10
	50	48.1	47.2	46.9	94.8	1.9–3
	100	113.6	104.8	115.7	111.4	4.8–14
Method 2	10	8.5	8.1	9.2	86	8–19
	50	47.8	45.3	38.2	87.6	5–23.6
	100	84.3	89.2	76.3	83.3	10–23.7
Method 3	10	8.6	9.5	8.3	88	5–8
	50	43.2	52.6	42.1	91.9	5–15.8
	100	85.6	84.8	94.7	88.3	5–14.4

## Data Availability

Not applicable.

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
