# Peer review of "The Effects of Different Antigen–Antibody Pairs on the Results of 20 Min ELISA and 8 Min Chromatographic Paper Test for Quantitative Detection of Acetamiprid in Vegetables"

_biosensors, 2022, doi:10.3390/bios12090730_

Round 1

Reviewer 1 Report

The manuscript biosensors-1875556 entitled “The effects of different antigen–antibody pairs on the results of 20-min ELISA and 8-min chromatographic-paper test for quantitative detection of acetamiprid in vegetables” described different approaches regarding acetamiprid immunoassay development. The developed methods have been analysed using ELISA assays and paper-based strip QD fluorescence assay. Among the parameters optimized for both assays development we find in the manuscript the coating antigen for acetamiprid (using a competitive immunoassay configuration). The cross reaction of developed immunoassay is tested with eight different acetamiprid similar compounds. Furthermore, this study compared three simple sample pre-treatment methods for analysing acetamiprid in vegetables using both immunoassays.

The paper is well written, and the results are quite interesting for immunoassay development and well-described. I consider the paper is well-enough to be published in Biosensors.

Some minor point to have in consideration are:

·        In Table 1, all organic formulates should be in the same format in order to have a clear vision of the structural differences among the Acetamiprid similar compounds.

·        More details about QD reader parameters can clarified detection steps.

·        Figure 4 caption should indicate which letter correspond to each coating agent. Furthermore, figure resolution should be improved.

Author Response

Dear  Reviewer:

Thank you for your letter and for the reviewer's comments  

concerning our manuscript( ID: biosensors-1875556).Those comments are all valuable and very helpful for revising and improving our paper, as well as the important guiding significance to our researches. We have studied comments carefully and have

made correction which we hope meet with approval. Revised

portion are marked in red in the manuscript.  The  response 

to the reviewer's comments are as flowing:

Response to the reviewer's comments:

Reviewer #1:

1.In Table 1, all organic formulates should be in the same format in order to have a clear vision of the structural differences among the Acetamiprid similar compounds.

Response: We are very sorry and we have replaced  the same format with clear vision of the structural differences . (Annotation for Reviewer #1-1)

  1. More details about QD reader parameters can clarified detection steps.

Response: Detection steps were clarified  according to the Reviewer's comments. ï¼ˆAnnotation for Reviewer #1-2)

 3.Figure 3 caption should indicate which letter correspond to each coating agent. Furthermore, figure resolution should be improved.

Response: We are very sorry and we have replaced  Fig.3(E)

 that meets the requirements in the manuscript and we have indicated which letter correspond to each coating agent for Figure 3 caption.(Annotation for Reviewer #1-3)

Special thanks to Reviewer #1 for your good comments.

Reviewer 2 Report

Acetamiprid (ACP) is a new type of insecticide with a wide range of applications and has certain mite killing activity. ACP can act on the nicotine acetylcholine receptor of the insect nervous system, interfere with the stimulation conduction of the insect nervous system, which leads to insect paralysis and eventually die. ACP is particularly effective on pests that have developed high resistance to organophosphorus, carbamate, and pyrethroid insecticides. In this manuscript, authors developed an 8-min QD-based immunochromatographic stripe for the rapid and accurate quantitative determination of ACP residues in vegetables, which is a very useful supplement to the current Chromatogram method. However, the research work in this article mainly focuses on the preparation of anti-ACP mAb and the measurement of antibody-antibody affinity. The design ideas, detection principles, etc. of immunochromatographic-strip are less. Judging from the entire manuscript, its innovation point does not seem to be in the biosensors. Suggestions for revision are as follows:

ABSTRACT:From the perspective of methodology, the abstract should simplify the monoclonal control process, focusing on the background of the research (existing problems), the methods used to solve the problem, and the main results and research significance.

INTRODUCTION: The author has fully explained the background and detection status of ACP. However, the third paragraph is concentrated in the search for high-affinity antigen-antibody pairs. There is no reflection of the innovation of immunological layer analysis note design.

“2. Material and methods” and “3. Methods”, the title of the section is repeated.

RESULTS:section 4.2, 6th line, “1 mg/m” ?

Author Response

Dear Editors and Reviewers:

Thank you for your letter and for the reviewer's comments  

concerning our manuscript( ID: biosensors-1875556).Those comments are all valuable and very helpful for revising and improving our paper, as well as the important guiding significance to our researches. We have studied comments carefully and have

made correction which we hope meet with approval. Revised

portion are marked in red in the manuscript.  The  response 

to the reviewer's comments are as flowing:

Reviewer #2:

Comment 1: ABSTRACT:From the perspective of methodology, the abstract should simplify the monoclonal control process, focusing on the background of the research (existing problems), the methods used to solve the problem, and the main results and research significance.

Response: We have simplified the antibody preparation process of the abstract, shifting the focus to the research context, problem-solving methods, the main results, and research implications . (Annotation for Reviewer #2-1)

Comment 2:INTRODUCTION: The author has fully explained the background and detection status of ACP. However, the third paragraph is concentrated in the search for high-affinity antigen-antibody pairs. There is no reflection of the innovation of immunological layer analysis note design.

Response:  We are very sorry and we have made correction according to the Reviewer's comments. ï¼ˆAnnotation for Reviewer #2-2)

Comment 3:  Material and methods” and “3. Methods”, the title of the section is repeated.RESULTS:section 4.2, 6th line, “1 mg/m” ?

Response:   We are very sorry and we have made correction .(Annotation for Reviewer #2-3)

Special thanks to Reviewer #2 for your good comments.

Round 2

Reviewer 2 Report

The revised version can be published in biosensors journal.